# Dwarf Pomegranate (*Punica granatum* L. var. *nana*): Source of 5-HMF and Bioactive Compounds with Applications in the Protection of Woody Crops

**DOI:** 10.3390/plants11040550

**Published:** 2022-02-18

**Authors:** Eva Sánchez-Hernández, Laura Buzón-Durán, José A. Cuchí-Oterino, Jesús Martín-Gil, Belén Lorenzo-Vidal, Pablo Martín-Ramos

**Affiliations:** 1Department of Agricultural and Forestry Engineering, ETSIIAA, University of Valladolid, Avenida de Madrid 44, 34004 Palencia, Spain; eva.sanchez.hernandez@uva.es (E.S.-H.); laura.buzon.duran@gmail.com (L.B.-D.); mgil@iaf.uva.es (J.M.-G.); 2Instituto Universitario de Investigación en Ingeniería de Aragón (i3A), EPS, University of Zaragoza, Carretera de Cuarte s/n, 22071 Huesca, Spain; cuchi@unizar.es; 3Servicio de Microbiología, Hospital Universitario Rio Hortega, Calle Dulzaina 2, 47012 Valladolid, Spain; blorenzov@saludcastillayleon.es; 4Instituto Universitario de Investigación en Ciencias Ambientales de Aragón (IUCA), EPS, University of Zaragoza, Carretera de Cuarte s/n, 22071 Huesca, Spain

**Keywords:** 5-hydroxymethylfurfural, dwarf pomegranate, FTIR, GC−MS, phytochemicals, pyrogallol, sitosterol

## Abstract

While the properties of edible pomegranate varieties have been widely explored, there is little information on ornamental types. In this study, possible alternatives for the valorization of dwarf pomegranate fruits have been explored. The characterization of their hydromethanolic extract by gas chromatography−mass spectrometry evidenced the presence of high contents of 5-hydroxymethylfurfural (a carbon-neutral feedstock for the production of fuels and other chemicals) and *β*- and *γ*-sitosterol stereoisomers. The microbicidal activity of the crude extract, both alone and in a conjugate complex with chitosan oligomers (COS), was investigated against three plant pathogenic microorganisms that cause significant losses in woody crops: *Erwinia amylovora*, *E. vitivora,* and *Diplodia seriata*. In in vitro assays, a strong synergistic behavior was found after conjugation of the bioactive constituents of the fruit extract with COS, resulting in minimum inhibitory concentration (MIC) values of 750 and 375 μg·mL^−1^ against *E. amylovora* and *E. vitivora*, respectively, and an EC_90_ value of 993 μg·mL^−1^ against *D. seriata*. Hence, extracts from the non-edible fruits of this *Punicaceae* may hold promise as a source of high value-added phytochemicals or as environmentally friendly agrochemicals.

## 1. Introduction

The pomegranate tree (*Punica granatum* L.), of the Lythraceae family, is a shrub native to the Middle East and the Mediterranean but is widely cultivated in warm-temperate regions around the world. The fruit from the edible varieties is the well-known pomegranate. A high genetic diversity of morphological and biochemical traits characterizes this species, with more than 500 cultivars identified worldwide [1].

The dwarf pomegranate (usually referred to as *P. granatum* var. *nana*, but also as *Punica granatum* ‘Nana’, *Punica granatum nana*, *P. granatum* var. *nana* Pers., and *Punica nana* L.) is a natural variant of the species. Even though it has sometimes been treated as the third species of *Punica* [2], in the work by Currò et al. [3] on microsatellite loci for pomegranate, it was evidenced that it did not show unique allele patterns.

This ornamental variety differs from the edible variety in its much smaller size, its lustrous, lanceolate leaves (ca. 2.5 cm long), and its small (ca. 5 cm) fruits. However, the shrub itself is not so dwarf, and after pruning, it usually stands between 60 and 90 cm high. In warm-temperate climates, it is deciduous, but indoors and in the tropics, it behaves as an evergreen. Its flowers are orange-red, with crinkled petals, and so are the calyx and ovary. The maturation of the fruits is slow, and they remain on the plant for more than six months. They are completely red when ripe, but are not edible, as they do not have the sweet flavor of the standard pomegranate [4].

All parts of edible varieties of pomegranate have been reported to have antioxidant properties [5], and their antimicrobial activity has been the subject of several recent review papers [6,7,8]. Regarding their potential as a source of bioactive products against phytopathogens, there are few examples of their antibacterial action (e.g., against *Pseudomonas syringae* pv. *tomato*, which causes the bacterial speck of tomato [9]), but extracts from the fruit (mainly those from the peel) have been widely used as antifungal agents: against *Alternaria alternata*, *A. solani*, *Botrytis cinerea*, *Colletotrichum* spp., *Fusarium* spp., *Monilia* spp., *Penicillium italicum*, *Pythium* spp., *Rhizoctonia solani*, *Rhizopus stolonifera,* and in the control of *Sclerotinia sclerotiorum* [10,11,12,13,14], among others. In fact, among the 24 plant extracts tested by Rongai et al. [15], the authors highlighted that pomegranate extracts showed the highest inhibition against *F. oxysporum* f.sp. *lycopersici*. However, it is known that antimicrobial activity is highly dependent on the cultivar [16].

Regarding the dwarf pomegranate tree, only limited studies have been carried out: El-Moghazy et al. [4] studied the macro- and micromorphology of its leaves, flowers, and stem bark. The same group also investigated the anti-inflammatory and antioxidant activities of the alcoholic extract of its leaves [17]. In addition, Emam et al. [18] studied the methanolic extract of the leaves, testing its nematicidal and fungicidal activity (against *R. solani, F. oxysorum* f.sp. *lycopersici,* and *Sclerotium rolfsii*). The antimicrobial effect of peel extracts was investigated against *Salmonella enterica* by Wafa et al. [19], and the ethyl acetate fraction of the bark was explored against hyperglycemia by El Deeb et al. [20].

Taking into consideration that at present the fruit of *P. granatum* var. *nana* has no application (beyond its ornamental function), the aim of this article is to perform a physicochemical characterization of whole fruits of *P. granatum* var. *nana* with a view to their valorization, exploring their potential as a bioenergy feedstock and as a source of environmentally friendly agrochemicals. In this sense, EU regulations (Council Regulation (EC) 834/2007, Commission Regulation (EC) 889/2008, Article 14 of Directive 2009/128/EC, Regulation (EU) 2019/1009, etc.) promote the use of products based on natural compounds for integrated pest control. Therefore, in this work, the efficacy of whole-fruit extracts of dwarf pomegranate has been investigated against three phytopathogenic microorganisms that cause important losses in woody crops: two bacteria, namely, *Erwinia amylovora* (Burrill) and *Erwinia vitivora* Du Plessis [synonym of *Xylophilus ampelinus* (Panagopoulos) Willems, Gillis, Kersters, van den Broeke & De Ley, also described as *Bacillus vitivorus* by Baccarini in 1893, and as *Xanthomonas ampelina* by Panagopoulos in 1969] [21], and a fungus, *Diplodia seriata* De Not. While *E. amylovora* and *E. vitivora* are the causal agents of fire blight in apple trees and bacterial necrosis of grapevine, respectively, *D. seriata* is the pathogen responsible for canker, black rot, and frogeye leaf spot in apple trees and is implicated in grapevine dead arm disease.

## 2. Results

### 2.1. Elementary Analysis

The percentages of C, H, and N of *P. granatum* var. *nana* fruits (in dry matter wt%) are summarized in Table 1. The moisture content was 5.67%, and structural water (recorded by thermal analysis) was 10.79%.

The higher heating value (HHV) derived from the elemental analysis data was 17.1 kJ·g^−1^.

### 2.2. Thermal Analysis

Thermogravimetry (TG), derivative thermogravimetry (DTG) and differential scanning calorimetry (DSC) curves for *P. granatum* var. *nana* fruits are shown in Appendix A. The exothermal effects that occur at 325 and 445 °C are due to holocellulose and lignin decomposition. The ash content at 600 °C was 4.4%. 

### 2.3. Vibrational Characterization

The main infrared absorption bands in the spectrum of *P. granatum* L. var. *nana* fruits (dried and ground), together with their assignments, are shown in Table 2. The most prominent ones appeared at 1730, 1444, 1150, and 1018 cm^−1^. The peak at 1730 cm^−1^ is attributed to unconjugated carbonyl stretching vibration in hemicelluloses (C−O stretching vibration in carboxyl, carbonyl, and acetyl groups); the peak at 1444 cm^−1^ to asymmetric C−H deformations; the peak at 1150 cm^−1^ to C−O−C vibrations in cellulose; and the peak at 1018 cm^−1^ to pectins or carotenes.

The infrared spectrum of the freeze-dried hydromethanolic extract showed bands at 3272, 2936, 2917, 2849, 1724, 1624, 1456, 1371, 1315, 1260, 1192, 1076, 1048, 1018, 951, 926, 876, 715, 652, 627, 518, 495, and 475 cm^−1^.

### 2.4. Analysis of the Constituents of the Fruit Extract by GC−MS

Gas chromatography−mass spectrometry (GC−MS) analysis of the hydromethanolic fruit extract (Table 3, Appendix A) allowed for the identification of 5-hydroxymethylfurfural (5-HMF, 37%); 2,3-dihydro-3,5-dihydroxy-6-methyl-(4H)-pyran-4-one (DDMP, 9.7%); *β*-sitosterol (7.2%); and 1,2,3-benzenetriol (pyrogallol, 6.1%) as the most prominent components (Figure 1). Palmitic and octadecadienoic acids, present in small amounts (1.2% and 1.1%, respectively), were also detected in *P. granatum* var. *nana* seed oil by Amri et al. [22], although in higher concentrations (7.8% and 2.7%, respectively).

### 2.5. Antimicrobial Activity

The activity of the hydromethanolic extract of *P. granatum* var. *nana* fruits and its three main constituents was tested alone and after conjugation with chitosan oligomers (COS). The results against the two quarantine bacterial plant pathogens are summarized in Table 4. It can be seen that the activity of the extract was comparable to that of COS alone, *β*-sitosterol, and 5-HMF, reaching total inhibition of the two *Erwinia* spp. at 1500 μg·mL^−1^ (whereas DDMP led to full inhibition at half that dose). However, after conjugation with COS, a marked potentiation of the antibacterial action of the extracts was found, with minimum inhibitory concentration (MIC) values of 375 and 250 μg·mL^−1^ against *E. amylovora* and *E. vitivora*, respectively, similar to those recorded for COS−DDMP and better than those found for COS−*β*-sitosterol and COS−5-HMF. 

MIC values for four conventional antibiotics (viz. amikacin, gentamicin, benzylpenicillin, and tetracycline), determined using ETEST^®^ strips (Appendix A), are provided in Appendix A for comparison purposes.

As for the antifungal behavior, tested against *D. seriata* (Figure 2 and Appendix A), the activity of the pure extracts was lower than that of chitosan, 5-HMF, and DDMP (for which total inhibition was achieved at 1500, 1000, and 750 μg·mL^−1^, respectively) and much lower than that of *β*-sitosterol (for which mycelial growth was completely inhibited at 250 μg·mL^−1^). The activity again increased after conjugation with COS, resulting in total inhibition at 1000 μg·mL^−1^. This inhibitory concentration was lower than that recorded for COS alone, which points to a synergistic behavior (see Table 5 and Table 6 in which the effective concentrations and synergistic factors are summarized, respectively). However, the best results corresponded to the COS−*β*-sitosterol conjugate complex, for which total inhibition was achieved at 187.5 μg·mL^−1^, followed by COS−DDMP and COS−5-HMF.

## 3. Discussion

### 3.1. Elemental Analysis

As for elemental analysis, the results obtained (Table 1) were close to those reported for *P. granatum* husk (C, 43.9%; H, 4.7%; N, 1.2%; S, 0.6%, according to Ömeroğlu Ay et al. [23]; and C, 42.9%; H, 4.1%; N, 1.3%, according to Bretanha et al. [24]). The C:N ratio (28.1) was lower than that reported for pomegranate peel (36.6–39.1 [23,25]). 

The moisture content (5.67%) was intermediate between those previously reported for pomegranate peel (13.7%) [26] and pomegranate seeds (5.82% according to Rowayshed et al. [26], and 6.84% according to Abiola et al. [27]). 

The calorific value derived from elemental analysis data (17.1 kJ·g^−1^) was higher than that reported for pomegranate peel (15.2 kJ·g^−1^) [28], but would still not meet the requirements of ISO 17225–2:2014 [29]/EN*plus* [30] (HHV ≥ 18.82 kJ·g^−1^) for valorization as a fuel.

### 3.2. Thermal Analysis

The ash content at 600 °C for fruits of the *nana* variety (4.4%) was intermediate between those reported for seeds and the peel of other pomegranate varieties/cultivars (with ash contents as low as 1.5% for the seeds [26,27] and 6.8% for the peel [28]).

### 3.3. Vibrational Characterization

The bands present in the freeze-dried hydromethanolic extract were consistent with the presence of the main constituents identified by GC−MS: the band at 2849 cm^−1^ can be assigned to 5-hydroxymethylfurfural (C−H vibration of the aldehyde group), that at 1048 cm^−1^ (C−O strain) to *β*-sitosterol, and those at 1360, 1311, 1246, 1184, 1065 and 702 cm^−1^ to pyrogallol (with a 10 cm^−1^ shift) [31].

### 3.4. Phytoconstituents Identified by GC−MS

Taking into consideration that the hydromethanolic extraction mixture also solubilizes polar compounds (non-volatile) that cannot be detected by GC−MS without previous derivatization of the extract, a word of caution concerning the results seems necessary. It should be clarified that, in the work presented herein, such prior derivatization was not conducted because it has a number of drawbacks: it makes procedural preparation steps longer and costlier (which would have a negative impact on the economic viability of the crop protection treatments), the data acquisition process becomes more complex and longer because derivatization can sometimes lead to impurities, notwithstanding the uncertainty of conversion of compounds into derivatives and the use of toxic reagents [32]. On the other hand, the injection of non-volatile compounds may result in eventual damage to the GC capillary column.

Upon comparison of the obtained results with other *P. granatum* extracts reported in the literature, it is worth noting that hydroxymethylfurfural and pyrogallol were also identified by Kumar and Vijayalakshmi [33] in an ethanolic extract of *P. granatum*, and similarities were also found with the phytoconstituents reported by Bonzanini et al. [34] (also in ethanolic extract): nitroisobutylglycerol, ethyl-*α*-D-glucopyranoside, 3-hydroxy-2-methyl-4H-pyran-4-one (maltol), 2,3-dihydro-3,5-dihydroxy-6-methyl-(4H)-pyran-4-one (or hydroxydihydromaltol), and 3H-indole-3-carbaldehyde(4-amino-5-methyl-4H-1,2,4 triazol-3-yl)hydrazone.

Concerning the high concentration of 5-HMF in the extract (37% peak area), it was initially ascribed to a failure in the extraction operating conditions (unexpected rise of the temperature), but the bibliographic reports of a 21% (peak area) content in an ethanolic extract by Kumar and Vijayalakshmi [33], a 32.1% (peak area) content by Hamad et al. [11], and a 39.7% content by Mohamad and Khalil [35] encouraged us to repeat the extraction and determination, verifying that the result was consistent. This result is also in agreement with a study by Fischer et al. [36], who explored the thermal impact on the anthocyanin and phenolic co-pigments of pomegranate juices and on the formation of 5-HMF and other degradation products and concluded that upon forced heating at 90 °C for five hours, the 5-HMF contents increased only slightly. The finding of such a high 5-HMF content is important because it is a chemical used as a flavoring agent in the food industry and industrially as a carbon-neutral feedstock for the production of fuels and other chemicals [37,38]. Further, it has antibacterial properties [39]: for instance, Kaur et al. [40] assayed the antibacterial activity of 5-HMF, and its derivatives from pomegranate fruits were tested for their antimicrobial potential against *Klebsiella* sp., finding MICs in the 40–160 μg·mL^−1^ range.

Among the other main constituents, 2,3-dihydro-3,5-dihydroxy-6-methyl-(4H)-pyran-4-one has preconized anti-cancer and antioxidant properties [41]; pyrogallol, a benzenetriol, has antiseptic activity against *S. aureus* and *E. coli* [42]; *β*-sitosterol has antibacterial activity similar to that of pyrogallol [43]); and *γ*-sitosterol is a potent inhibitor of the complement component C1 complex, with potential anticancer activity [44].

### 3.5. Comparison of the Microbicidal Activity of the Extract

Reports on the antimicrobial activity of *P. granatum* var. *nana* are scarce in the literature: the antibacterial activity of dwarf pomegranate peel extracts (in water, aqueous ethanol, and aqueous ethanol−methanol) was tested against *Salmonella enterica* by Wafa et al. [19], with moderate results, and the hydromethanolic extract of *P. granatum* var. *nana* leaves was tested against *S. rolfsii*, *R. solani*, and *F. oxysporum* f.sp. *lycopersici* by Emam et al. [18]. To the best of the authors’ knowledge, no studies on the activity of fruit extracts have been published to date.

Conversely, the microbicidal activity of edible pomegranate varieties has received more attention and has been recently addressed in review articles by Singh et al. [6], Pirzadeh et al. [7], and Chen et al. [8]. Since no reports are available against the plant pathogens studied here, results against other *Erwinia* spp., *Xanthomonas* spp. (since *E. vitivora* is also known as *Xanthomonas ampelina*), and Botryosphaeriaceae are presented for comparison purposes.

Hassan et al. [45] investigated the antibacterial activity of ethanolic extracts of the peel of 32 Egyptian pomegranate cultivars against *E. carotovora* and *X. campestris*, finding diameter zone of inhibition (DZI) values in the 10 to 37.5 mm range and in the 12.8 to 36.6 mm range, respectively, depending on the *P. granatum* cultivar. Mhaisgawali et al. [46] assessed the antibacterial activity of pomegranate bark against *E. chrysanthemi* and *X. malvacearum*, finding efficacies in terms of DZI values similar to those of streptomycin. Truchado et al. [47] reported that pomegranate extract at 20 μg·mL^−1^ caused significant inhibition of *E. carotovora* quorum-sensing signals, and Vlachou et al. [48] found that pomegranate extracts showed good control of *E. carotovora* pv. *atroseptica* and *Xanthomonas* spp. at a concentration of 1 mg/disc. Hussein et al. [49] tested five plant extracts in dimethyl sulfoxide, including *P. granatum* peel, against *E. amylovora*, concluding that the inhibition was greater than that of black cumin and thyme. However, none of these studies reported MIC values. The only study in which such information was available was the one by Khaleel et al. [50], who tested the efficacy of the ethyl acetate extract of *P. granatum* peel against *E. carotovorum* and *X. gardneri*, finding MIC values of 3.125 and 6.25 mg·mL^−1^, respectively. In comparison, the hydromethanolic extracts of *P. granatum* var. *nana* fruits tested here were 2 to 4 times more effective (MIC = 1.5 mg·mL^−1^). Nonetheless, in view of the MIC values presented in Appendix A, their effectiveness is still much lower than that of conventional antibiotics (for which MIC values less than 2 μg·mL^−1^ were obtained against *E. amylovora* and *E. vitivora*). However, it should be clarified that presently there are no antibiotics authorized as plant protection products in the EU. Some EU member states authorize their emergency use to control outbreaks, but the volumes used are negligible, and their application is strictly controlled [51].

Regarding the antifungal activity against Botryosphaeriaceae fungi, Matos et al. [52] observed no inhibition of *Lasiodiplodia theobromae* for pomegranate oil at concentrations of up to 0.6%, but Lorenzetti et al. [53] reported a significant inhibition (87%) of *Diplodia macrospora* at a concentration of 7.4% (i.e., at 74,000 μg·mL^−1^) of pomegranate peel extract. In comparison, the non-conjugated dwarf pomegranate fruit extract, for which an EC_90_ of 4600 μg·mL^−1^ was estimated, should be much more effective (by a factor of 16).

### 3.6. On the Synergistic Behavior after Conjugation with Chitosan Oligomers

Although they do not refer to conjugated complexes, but to compounds, there are previous reports in the literature of improved antimicrobial activity by adding pomegranate extracts to chitosan coatings: for example, incorporation of pomegranate peel extracts in water and methanol into chitosan resulted in a significant reduction of the fungal growth and activity of *Penicillium* spp. [54,55], finding that chitosan enhanced the action of the plant extracts to inhibit pathogenic fungi. The incorporation of peel extracts into chitosan coatings also inhibited microbial growth, prolonged the shelf life, and maintained the sensory scores of pepper [56]. 

An explanation of the mechanism of action behind the enhanced antifungal effect in comparison with the use of individual natural agents has not been reported. According to Tayel et al. [55], the synergism can be tentatively explained by the various fungicidal components of each agent applied and by the fact that fungal pathogens are not readily resistant to multiple fungitoxicants. An alternative explanation would be based on enhanced solubility and bioavailability as a result of increased binding to specific negatively charged binding receptors on bacterial and fungal membranes.

## 4. Material and Methods

### 4.1. Plant Material

The 10-year-old specimens of *P. granatum* var. *nana* under study were purchased at one of the flower market stalls in the Plaza Bib-Rambla (also named ‘Plaza de las Flores’) in Granada (Spain). After one year in a pot, they were planted in Rales (Asturias, Spain). A voucher specimen, identified and authenticated by Dr. P. Pablo Ferrer-Gallego, was deposited at the VAL (Herbarium of the Botanical Garden of the University of Valencia, Valencia, Spain), code VAL244427. 

In 2020, 15 individuals were harvested to obtain fruit composite samples for analysis. The samples were shade dried and pulverized to a fine powder in a mechanical grinder. The moisture content was measured by weight loss after heating at 105 °C until a constant weight was reached.

### 4.2. Reagents

High-molecular-weight chitosan (CAS 9012-76-4) was supplied by Hangzhou Simit Chem. & Tech. Co. (Hangzhou, China). Neutrase^TM^ 0.8 L enzyme was acquired from Novozymes A/S (Bagsværd, Denmark). Chitosan oligomers with MW < 2000 Da were obtained following the procedure described by Santos-Moriano et al. [57], with the modifications reported in [58]. *β*-sitosterol (CAS 83-46-5, analytical standard), 3-hydroxy-2-methyl-4-pyrone (CAS 118-71-8), 5-(hydroxymethyl)furfural (CAS 67-47-0), methanol (CAS 67-56-1), tryptic soy agar (TSA, CAS 91079-40-2), and tryptic soy broth (TSB, CAS 8013-01-2) were supplied by Sigma–Aldrich Química (Madrid, Spain). Potato dextrose agar (PDA) was purchased from Becton Dickinson (Bergen County, NJ, USA).

### 4.3. Phytopathogen Isolates

*E. amylovora* (NCPPB 595) and *E. vitivora* (CCUG 21976) strains used in the study were obtained from the Spanish Type Culture Collection (CECT; Valencia, Spain). The *D. seriata* isolate (Y-084-01-01a) was supplied as a lyophilized vial (later reconstituted and refreshed as a subculture in PDA) by Instituto Tecnológico Agrario de Castilla y León (ITACYL; Valladolid, Spain).

### 4.4. Preparation of the Fruis Extract

The *P. granatum* var. *nana* fruit sample was mixed (1:20, *w/v*) with a methanol/water solution (1:1 *v/v*). The solution was heated in a water bath at 50 °C for 30 min and sonicated for 5 min (2.5 min−1 min stop−2.5 min) using a 1000 W, 20 kHz probe-type ultrasonicator (model UIP1000hdT; Hielscher Ultrasonics, Teltow, Germany). After centrifugation at 9000 rpm for 15 min, the supernatant was filtered using Whatman No. 1 paper. Aliquots of the final solution were freeze-dried for infrared vibrational analysis.

### 4.5. Plant Biomass and Fruit Extract Physicochemical Characterization

Elemental analysis (CHNS) of the fruits (once dried and ground) was carried out with a LECO CHNS-932 apparatus (St. Joseph, MI, USA). Calorific values were calculated from elemental analysis data by using the following equation: HHV = (0.341 × %C) + (1.322 × %H) − 0.12(%O + %N), where HHV is the higher heating value, i.e., the calorific value of the dry material expressed in kJ·g^−1^, and %C, %H, %O, and %N represent the mass fractions (in wt% of dry material) [59].

Thermogravimetry and differential scanning calorimetry analyses were performed using a simultaneous TG-DSC2 (Mettler Toledo; Columbus, OH, USA) in N_2_:O_2_ atmosphere (4:1) with a heating ramp of 20 °C·min^−1^. According to the usual pyrolysis temperature conditions in oxygen bomb calorimeters, the ash content was estimated from the residue obtained upon heating up to 600 °C [60]. 

The infrared vibrational spectra were recorded with a Nicolet iS50 Fourier-transform infrared spectrometer (Thermo Scientific; Waltham, MA, USA) equipped with an attenuated total reflection (ATR) system. Spectra were collected with a resolution of 1 cm^−1^ in the 400–4000 cm^−1^ range, taking the interferograms that resulted from the co-addition of 64 scans. The spectra were processed using the advanced ATR correction algorithm [61] available in OMNIC^TM^ software.

The hydroalcoholic fruit extract was studied by gas GC−MS at the STI facilities at the University of Alicante (Alicante, Spain), using a 7890A gas chromatograph coupled to a 5975C quadrupole mass spectrometer (both from Agilent Technologies; Santa Clara, CA, USA). The chromatographic conditions were: 3 injections/vial, 1 µL injection volume; 280 °C injector temperature, in splitless mode; 60 °C initial oven temperature, kept for 2 min, followed by a heating ramp of 10 °C·min^−1^ up to a final temperature of 300 °C, kept for 15 min. The chromatographic column used for the separation of the compounds was an HP-5MS UI (Agilent Technologies), with length = 30 m, diameter = 0.25 mm, and film thickness = 0.25 µm. Concerning the mass spectrometer conditions, the temperature of the electron impact source of the mass spectrometer was 230 °C, and that of the quadrupole was 150 °C; with a 70 eV ionization energy. For equipment calibration, test mixture 2 for apolar capillary columns according to Grob (Supelco 86501) and PFTBA tuning standards were used. The identification of the extract constituents was based on a comparison of their mass spectra and retention times with those of the authentic compounds and by computer matching with the National Institute of Standards and Techniques (NIST11) database and the monograph by Adams [62].

### 4.6. In Vitro Antimicrobial Activity Assessment

CLSI standard M07-11 [63] was followed for the determination of the antibacterial activity. An isolated colony of *E. amylovora* was first incubated at 30 °C for 18 h in TSB liquid medium. Starting from a 10^8^ colony forming units (CFU)·mL^−1^ concentration, serial dilutions were conducted to obtain a final inoculum concentration of ~10^4^ CFU·mL^−1^. The bacterial suspensions were then delivered to the surface of TSA plates, to which the products under investigation had previously been added at concentrations ranging from 62.5 to 1500 μg·mL^−1^. The incubation of the plates was carried out at 30 °C for 24 h. The same procedure was followed for *E. vitivora*, albeit at a temperature of 26 °C. Readings were taken after 24 h. The MIC values were determined as the lowest concentrations at which no bacterial growth was observed. The experiments were run in triplicate, with each replicate consisting of three plates per treatment and concentration combination.

To compare the effectiveness of the natural compounds with that of standard antibiotics, ETEST^®^ gradient MIC strips (bioMérieux Clinical Diagnostics, Marcy-l’Étoile, France) were chosen. Well-isolated colonies from agar plates were suspended in 0.85% NaCl suspension medium to obtain an inoculum turbidity of 0.5 McFarland; a sterile swab was soaked in the inoculum and, after removing excess fluid, it was used to carefully streak the entire surface of the Mueller–Hinton agar plates three times (rotating the plate 60° each time to evenly distribute the inoculum), allowing a drying time of 20 min; ETEST^®^ strips (with amikacin, benzylpenicillin, gentamicin, and tetracycline) were then positioned on the plates, using one strip per plate; plates were incubated overnight in an inverted position (lid down), and the MICs were read where the edge of the inhibition ellipse intersected the side of the strip.

As for the antifungal activity, it was assessed following EUCAST standard antifungal susceptibility testing procedures [64]. The agar dilution method was used in which aliquots of stock solutions were incorporated onto the PDA medium to obtain concentrations in the 62.5−1500 μg·mL^−1^ interval. Five-millimeter mycelial plugs from the margin of 1-week-old PDA cultures of *D. seriata* were transferred to plates that incorporated the aforementioned concentrations of each product (with three plates per treatment and concentration combination and two replicates). Incubation was conducted at 25 °C in the dark for seven days. PDA medium without any amendment was used as the control. Mycelial growth inhibition was calculated as $\left(\left(d_{c}-d_{t}\right)/d_{c}\right)\times 100$, where *d_c_* and *d_t_* are the average diameters of the fungal colony of the control and of the treated fungal colony, respectively. The 50% and 90% effective concentrations (EC_50_ and EC_90_, respectively) were calculated through PROBIT analysis in IBM SPSS Statistics v.25 (IBM; Armonk, NY, USA). Wadley’s method [65] was chosen to determine the level of interaction.

### 4.7. Statistical Analysis

After checking that homogeneity and homoscedasticity requirements were met (using Shapiro–Wilk and Levene tests, respectively), the statistical analysis of the mycelial growth inhibition results for *D. seriata* was carried out by analysis of variance (ANOVA), followed by post hoc comparison of means through Tukey’s test (*p* < 0.05), using IBM SPSS Statistics v.25.

## 5. Conclusions

Although the ash content of the inedible fruits of *P. granatum* var. *nana* impedes their use as biofuel, the finding by GC−MS in the hydroalcoholic extract of significant amounts (37%) of 5-hydroxymethylfurfural (key for the production of fuels and other chemicals), validates their usefulness as a promising energy feedstock. Moreover, due to their content of *β*-sitosterol (and other phytochemicals such as 2,3-dihydro-3,5-dihydroxy-6-methyl-(4H)-pyran-4-one), fruits of *P. granatum* var. *nana* can also be used as a source of agrochemicals for the control of fire blight and ‘maladie d’Oléron’, although conjugation with COS is required to promote the efficacy of the extracts, resulting in MIC values as low as 375 and 250 μg·mL^−1^ against *E. amylovora* and *E. vitivora*, respectively. The antifungal behavior of the conjugated complexes, tested against *D. seriata*, was moderate (EC_90_ = 993 μg·mL^−1^), but suggests that they may have a broader spectrum of antibiotic activity.

## Figures and Tables

**Figure 1 plants-11-00550-f001:**
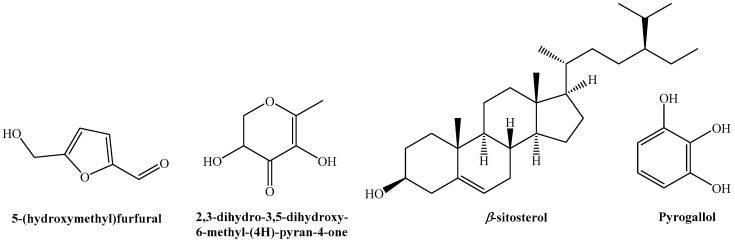
Main phytochemicals identified in the hydromethanolic extract of *P. granatum* var. *nana* fruits.

**Figure 2 plants-11-00550-f002:**
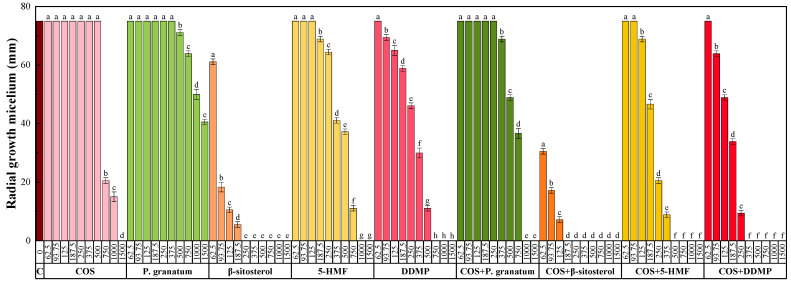
Radial growth of the mycelium of *D. seriata* in in vitro assays performed on PDA medium with different concentrations (in the 62.5–1500 μg·mL^−1^ range) of chitosan oligomers (COS), hydromethanolic extract of *P. granatum* var. *nana* fruits, the main phytochemical constituents of the extract, and their respective conjugated complexes. The same letters above concentrations mean that they are not significantly different at *p* < 0.05. Error bars represent standard deviations.

**Table 1 plants-11-00550-t001:** Elemental analysis of *P. granatum* L. var. *nana* fruits (after drying and grinding). Values are presented as the mean of four replicates, followed by minimum and maximum values in parentheses.

C (%)	H (%)	N (%)	O (by Difference, %)	C:N Ratio
43.15 (42.7–43.4)	6.41 (6.3–6.5)	1.54 (1.3–1.6)	48.9	28.1

**Table 2 plants-11-00550-t002:** Main bands in the infrared spectrum of *P. granatum* L. var. *nana* fruits.

WaveNumber (cm^−1^)	Assignment
3335	bonded O−H stretching (cellulose, hemicellulose, lignin)
2919	−CH_2_ asymmetric stretching (alkyls)
2850	−CH_2_ symmetric stretching (cutin)/CH_2_−(C6)−bending (cellulose)/ C−H vibration of the aldehyde group (5-HMF)
1730	C = O stretching (alkyl esters)
1624	C = O stretching (hemicellulose, bonded ketones, …)/C−C-stretching
1517	aromatic skeletal (aromatic carotenoids)
1444	C−H deformation/C = C stretching of furan ring (furfural)/O−CH_3_ stretching
1325	CH in-plane bending (celluloses I and II)
1226	C−C−O asymmetric stretching (acetylated glucomannan)/ C−O and OH of the COOH/amide III
1150	C−O−C asymmetric stretching (celluloses I and II)/C−C in-plane (*β*−carotene)
1101	C−O−C stretching (pyranose ring skeleton in cellulose)
1018	C−H bending (carotenes)/polygalacturonic acid (pectin present in plant cuticles)
913	*β*−glycosidic linkage
830	CH_2_ rocking deformation/O−C=O in-plane deformation

**Table 3 plants-11-00550-t003:** Main constituents of the hydromethanolic extract of *Punica granatum* var. *nana* fruits (only phytochemical compounds with peak areas above 1% are shown).

Peak	Retention Time (min)	Area (%)	Assignment
3	4.729	1.08	propanoic acid, 2-oxo-, methyl ester
8	5.752	1.06	2-furancarboxaldehyde, 5-methyl- (i.e., 5-methylfurfural)
15	7.115	1.17	hexanoic acid, 3-hydroxy-, methyl ester
18	7.548	1.19	2,5-furandicarboxaldehyde (i.e., 5-formylfurfural)
19	7.704	1.96	3,3-diacetyl-2,3,4,5-tetrahydro-2-oxofuran
23	8.716	7.89	4H-pyran-4-one, 2,3-dihydro-3,5-dihydroxy-6-methyl- (i.e., DDMP)
24	8.741	1.81	4H-pyran-4-one, 2,3-dihydro-3,5-dihydroxy-6-methyl- (i.e., DDMP)
31	10.231	37.00	5-hydroxymethylfurfural (i.e., 5-HMF)
32	10.314	2.28	oxiniacic acid
36	12.101	6.11	1,2,3-benzenetriol (i.e., pyrogallol)
37	12.317	4.23	hexanoic acid, 2-ethyl-
41	14.700	1.47	terpinen-4-ol
44	18.391	1.19	n-hexadecanoic acid (palmitic acid)
53	24.559	1.06	9,12-octadecadienoic acid (Z,Z)-
57	27.987	1.61	D-*α*-tocopherol
61	30.204	7.21	*β*-/*γ*-sitosterol

**Table 4 plants-11-00550-t004:** Antibacterial activity against *Erwinia* spp. of chitosan oligomers, *P. granatum* var. *nana* fruit hydromethanolic extract, its main bioactive constituents (*β*-sitosterol, 5-HMF, and DDMP), and their respective conjugate complexes.

Pathogen	Compound	Concentration (μg·mL^−1^)									
		62.5	93.75	125	187.5	250	375	500	750	1000	1500
*E. amylovora*	COS	+	+	+	+	+	+	+	+	+	−
	*P. granatum*	+	+	+	+	+	+	+	+	+	−
	*β*-sitosterol	+	+	+	+	+	+	+	+	+	−
	5-HMF	+	+	+	+	+	+	+	+	+	−
	DDMP	+	+	+	+	+	+	+	−	−	−
	COS−*P. granatum*	+	+	+	+	+	−	−	−	−	−
	COS−*β*-sitosterol	+	+	+	+	+	−	−	−	−	−
	COS−5-HMF	+	+	+	+	+	+	+	−	−	−
	COS−DDMP	+	+	+	+	+	−	−	−	−	−
*E. vitivora*	COS	+	+	+	+	+	+	+	+	+	−
	*P. granatum*	+	+	+	+	+	+	+	+	+	−
	*β*-sitosterol	+	+	+	+	+	+	+	+	+	−
	5-HMF	+	+	+	+	+	+	+	+	−	−
	DDMP	+	+	+	+	+	+	−	−	−	−
	COS−*P. granatum*	+	+	+	+	−	−	−	−	−	−
	COS−*β*-sitosterol	+	+	+	+	+	+	−	−	−	−
	COS−5-HMF	+	+	+	+	+	+	−	−	−	−
	COS−DDMP	+	+	+	+	−	−	−	−	−	−

COS = chitosan oligomers; 5-HMF = 5-hydroxymethylfurfural; and DDMP = 2,3-dihydro-3,5-dihydroxy-6-methyl-4H-pyran-4-one. “+” and “−“ indicate bacterial growth presence and absence, respectively.

**Table 5 plants-11-00550-t005:** Effective concentrations (expressed in μg·mL^−1^) against *D. seriata* of the hydromethanolic extract of *P. granatum* var. *nana* fruit and its three main constituents, alone and after conjugation with chitosan oligomers.

EC	COS	*P. granatum*	COS− *P. granatum*	*β*−Sitosterol	COS− *β*-sitosterol	5-HMF	COS− 5-HMF	DDMP	COS− DDMP
EC_50_	744.4	1656.4	623.0	82.0	51.0	442.6	212.8	317.8	158.0
EC_90_	1179.9	4639.6	992.8	151.2	124.4	847.9	394.4	699.3	314.0

COS = chitosan oligomers; 5-HMF = 5-hydroxymethylfurfural; DDMP = 2,3-dihydro-3,5-dihydroxy-6-methyl-4H-pyran-4-one.

**Table 6 plants-11-00550-t006:** Synergy factors for the conjugate complexes of COS with the hydromethanolic extract of *P. granatum* var. *nana* fruit and its three main constituents.

SF	COS−*P. granatum*	COS−*β*-Sitosterol	COS−5-HMF	COS−DDMP
EC_50_	1.65	2.90	2.61	2.82
EC_90_	1.89	2.15	2.50	2.80

## Data Availability

The data presented in this study are available on request from the corresponding author. The data are not publicly available due to their relevance to an ongoing Ph.D. thesis.

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
