# Peer review of "Dwarf Pomegranate (Punica granatum L. var. nana): Source of 5-HMF and Bioactive Compounds with Applications in the Protection of Woody Crops"

_plants, 2022, doi:10.3390/plants11040550_

Round 1

Reviewer 1 Report

This manuscript contains interesting data about phytochemical characteristics of dwarf pomegranate, but still needs to be improved before its acceptance. 

In general plant fruits, and especially pomegranate fruits, contain phenolic compounds.  Why authors did not analyze the content and composition of phenolics in this fruit? That data would be of high importance for the phytochemical characterization of this plant. Moreover, the extraction authors performed is normally suitable for the phenolic compounds, not for volatiles.  

As a minor correction, authors should cite tables properly. 

Reviewer 2 Report

Please refer to the observations available as notes in the enclosed file.

Reviewer 3 Report

The nomenclature of E. vitivora should be better defined, including the chronology of subsequent classifications in the text

Round 2

Reviewer 2 Report

The authors addressed all the previous comments and the manuscript has been deeply improved. This reviewer has no other observations.

Concerning the reply below:

"Q4. L111. GC is not the best analytic method to analyse hydromethanolic extracts since this 
extraction mixture solubilizes also several polar compounds (non-volatile) which cannot be 
detected without the previous derivatization of the extract and there is no evidence that this 
chemical modification to obtain more volatile derivatives has been performed. 
Response: We agree with the Reviewer that doubts may arise about the loss of some volatile 
derivatives, and -in this regard- we should clarify that we considered the possibility of following 
a derivatization procedure before carrying out the GC-MS analysis. Nonetheless, as the Reviewer 
should be aware of, derivatization has a number of drawbacks: it makes procedural preparation 
steps longer and costlier (which would have a negative impact on the economic viability of the 
crop protection treatments), the data acquisition process becomes more complex and longer 
because derivatization can sometimes lead to impurities, notwithstanding the uncertainty of 
conversion of compounds into derivatives and the use of toxic reagents [Journal of Food and Drug 
Analysis: Vol. 16 : Iss. 1 , Article 1]. Hence, we finally chose to conduct the GC-MS analysis 
without prior derivatization. Please kindly note that we have followed this approach before in 
articles published in this same journal (see, for instance, Plants 2021, 10(9), 1876; 
https://doi.org/10.3390/plants10091876), but -in any case- a word of caution has been included 
in the Discussion section (at the beginning of subsection 3.4) to ensure that possible limitations 
are clear to the reader."

I would only point out that injecting non volatile compounds in the GC capillary column this may be damaged.

Author Response

I would only point out that injecting non volatile compounds in the GC capillary column this may be damaged.

Response: We thank the Reviewer for bringing this possible problem associated with our methodology to our attention. A brief comment has been included in the manuscript in subsection 3.4, indicating that "[...] On the other hand, the injection of non-volatile compounds may result in damage to the GC capillary column." Please kindly note that, at present, we are outsourcing the GC-MS analyses at the STI facilities at the University of Alicante (Alicante, Spain) and -so far- they have not had any issues upon injection of the hydromethanolic extracts (although our samples only represent a very small percentage of those processed in these facilities). Nonetheless, we will use the derivatization approach in future analyses, as recommended by the Reviewer, to avoid eventual damage to the equipment.